# Targeting the Angiotensin II Type 1 Receptor in Cerebrovascular Diseases: Biased Signaling Raises New Hopes

**DOI:** 10.3390/ijms22136738

**Published:** 2021-06-23

**Authors:** Céline Delaitre, Michel Boisbrun, Sandra Lecat, François Dupuis

**Affiliations:** 1CITHEFOR, Université de Lorraine, F-54000 Nancy, France; celine.delaitre@univ-lorraine.fr; 2Biotechnologie et Signalisation Cellulaire, UMR7242 CNRS/Université de Strasbourg, 300 Boulevard Sébastien Brant, CS 10413, CEDEX, 67412 Illkirch-Graffenstaden, France; sandra.lecat@unistra.fr; 3CNRS, L2CM, Université de Lorraine, F-54000 Nancy, France; michel.boisbrun@univ-lorraine.fr

**Keywords:** AT_1_ receptor, Angiotensin II, cerebrovascular disease, biased agonism, beta-arrestin, RAS, TRV023, TRV027, Ang-(1–7)

## Abstract

The physiological and pathophysiological relevance of the angiotensin II type 1 (AT_1_) G protein-coupled receptor no longer needs to be proven in the cardiovascular system. The renin–angiotensin system and the AT_1_ receptor are the targets of several classes of therapeutics (such as angiotensin converting enzyme inhibitors or angiotensin receptor blockers, ARBs) used as first-line treatments in cardiovascular diseases. The importance of AT_1_ in the regulation of the cerebrovascular system is also acknowledged. However, despite numerous beneficial effects in preclinical experiments, ARBs do not induce satisfactory curative results in clinical stroke studies. A better understanding of AT_1_ signaling and the development of biased AT_1_ agonists, able to selectively activate the β-arrestin transduction pathway rather than the G_q_ pathway, have led to new therapeutic strategies to target detrimental effects of AT_1_ activation. In this paper, we review the involvement of AT_1_ in cerebrovascular diseases as well as recent advances in the understanding of its molecular dynamics and biased or non-biased signaling. We also describe why these alternative signaling pathways induced by β-arrestin biased AT_1_ agonists could be considered as new therapeutic avenues for cerebrovascular diseases.

## 1. Introduction

The systemic renin-angiotensin system (RAS) is a well-known hormonal cascade involved in many regulation processes such as the control of arterial pressure and vascular homeostasis. Angiotensinogen, a protein produced by the liver, is turned into angiotensin I, a decapeptide, by renin, an enzyme released by the kidney in response to various stimuli (blood pressure decrease, sympathetic stimulation). Angiotensin II (Ang II), an octapeptide, is generated by the cleavage of angiotensin I by the angiotensin-converting enzyme (ACE) mainly expressed at the endothelial surface. Ang II is known as the main effector of the RAS. Ang II binds with a similar affinity to two receptors: the Ang II type 1 receptor (AT_1_) and type 2 receptor (AT_2_), both belonging to the G protein-coupled receptor (GPCR) family and sharing 50% homology and 34% identity in their amino acid sequence [1,2,3].

Ang II activation of AT_1_ is acknowledged as triggering most of the known effects of RAS stimulation [4,5], such as vasoconstriction, water and sodium retention and aldosterone release by the adrenal glands. This leads to increases in blood pressure, cardiovascular remodeling and fibrosis [6,7,8]. In most physiological situations involving Ang II release, AT_1_ activation is predominant and will blunt the opposite effects induced by AT_2_ stimulation, such as vasodilation, apoptosis and antiproliferative effects [9,10,11,12]. Despite this AT_1_ predominance in Ang II mediated effects (mostly related to a more abundant expression of AT_1_ compared to AT_2_), the concept of a “protective arm” of RAS is a subject of intensive research: it is composed of AT_2_ and the Mas receptor (MasR), a GPCR activated by an endogenous truncated form of Ang II (Ang-(1–7)). For extensive reviews on the protective effect of AT_2_ activation, please refer to Matavelli’s or Carey’s works [13,14] and for the MasR, refer to Povlsen’s work [15].

Due to its wide physiological effects, AT_1_ plays a critical role in many pathological conditions and cardiovascular diseases, like cardiac hypertrophy, hypertension and heart failure. For decades, treatments preventing AT_1_ stimulation have been widely used as first line therapies for these cardiovascular diseases. ACE inhibitors, which prevent the cleavage of angiotensin I into Ang II, were developed in the 1980s. AT_1_ selective antagonists, called AT_1_ receptor blockers (ARBs or sartans), are a biphenyl-tetrazole based family of compounds first commercialized in the 1990s.

The involvement of AT_1_ in the regulation of cerebral circulation is also well established [16,17] and a cerebrovascular local production of Ang II has been shown [18]. AT_1_-dependent processes have been reported in cerebrovascular diseases including ischemic and hemorrhagic strokes, subarachnoid aneurysms, traumatic brain injury and might also be found in neurodegenerative processes, like Alzheimer’s and Parkinson’s diseases [19,20]. To date, etiological therapeutic treatments and/or therapeutic alternatives are scarce for all of these pathological conditions. This lack of efficient therapeutic options further justifies the growing interest for RAS and AT_1_ as a therapeutic target in brain disorders.

AT_1_ signaling pathways have been extensively studied over the past 25 years. AT_1_ is coupled to the heterotrimeric G_q/11_ protein that represents the main AT_1_ signal transduction process involved in vasoconstriction of vascular smooth muscle cells (VSMC). However, recent advances suggest that AT_1_ activation also leads to β-arrestin-dependent signaling pathways that could exert protective effects against cardiovascular diseases.

In the present review, we first discuss the major role of AT_1_ in cerebrovascular diseases, before reviewing the preclinical and clinical studies showing the interest and limits of AT_1_ blockade with ARBs in such situations (with a particular emphasis on strokes). We analyze the structure/function relationship of AT_1_ with different agonists based on recent seminal articles that have led to the demonstration that a specific alternative active conformational state of AT_1_ almost exclusively activates β-arrestin signaling instead of G protein signaling. Finally, we show the therapeutic potential of promoting this exclusive AT_1_/β-arrestin biased signaling in pathophysiological situations where cerebral circulation is altered.

## 2. Implication of AT_1_ in Cerebrovascular Disease

### 2.1. Role of AT_1_ Activation in Strokes

Strokes are a major cause of death in industrialized countries and may lead to various physical disabilities consecutive to neurological deficit [21]. Strokes may result from two distinct etiologies: ischemia or hemorrhage. An ischemic stroke is the consequence of an obstruction or a vasospasm of a cerebral artery or arteriole [22], whereas a hemorrhagic stroke results from the rupture of the arterial, or arteriolar, wall. In both cases, the subsequent blood deprivation will lead to reduced oxygen levels, ischemia and necrosis of the downstream brain tissue.

Numerous preclinical data suggest that an overactivation of AT_1_ by Ang II might contribute to the severity of cerebral ischemia and its consequences, mainly through its vasoconstrictor effects on cerebral arteries [23]. Indeed, in a classic rodent model of ischemic stroke, the middle cerebral artery occlusion model (MCAo, in which the middle cerebral artery is temporarily occluded by a nylon monofilament), increases in the local production of Ang II and in the AT_1_-dependent vasoconstriction of cerebral arteries have been reported [24,25]. In the context of cerebral ischemia, Ang II and AT_1_ stimulation thus lead to a decreased perfusion of the penumbra and might expand the deleterious consequences of a stroke [26]. Besides these acute post-stroke effects, chronic AT_1_ stimulation is the main determinant of hypertension-induced inward remodeling of the cerebral arteriolar wall [17,27,28]. Indeed, hypertension, one of the major risk factors for strokes, induces a structural narrowing of the cerebral arterial lumen (thus reducing blood perfusion in case of a stroke) that only antihypertensive treatments targeting RAS and AT_1_ stimulation (ACE inhibitors and/or ARBs) are able to prevent [17,28,29].

Ang II and AT_1_ stimulation also induce a proinflammatory response [30] and increase oxidative stress through the production of ROS, thus promoting cellular damage and apoptosis [31,32]. These effects might also contribute to the increase in brain damage and the enlargement of the penumbra in case of ischemia and during reperfusion.

Numerous preclinical and clinical studies have investigated the effects of ARBs, confirming the involvement of AT_1_ in cerebral ischemia. These studies have been reviewed recently [33] and are discussed in Section 3.

### 2.2. Role of AT_1_ Activation in Cerebral Aneurysm

Cerebral aneurysms are an abnormal condition in which a weakness of the cerebral arterial wall causes a localized dilation or bulge. Cerebral aneurysms can thus lead to hemorrhagic strokes showing a high mortality rate [34]. Given their role as major determinants of the cerebral arterial and arteriolar wall structure, RAS and AT_1_ might be good candidates to participate in the development of cerebral aneurysms.

Regarding abdominal aortic aneurysms, the involvement of Ang II and RAS stimulation has been clearly demonstrated in animals, and strong evidence have been gathered in humans [35,36]. However, when considering cerebral aneurysms, the role of RAS components and AT_1_ seems less obvious. In a study comparing patients with unruptured or ruptured cerebral aneurysm to controls, a significant drop in the mRNA expressions of ACE and AT_1_ has been observed [37]. Immunohistochemistry also revealed a decrease in AT_1_ and Ang II expression in unruptured aneurysmal walls. This decrease was even more pronounced in the arterial walls of patients with ruptured cerebral aneurysm [37]. Given the hypertrophic and remodeling effect of AT_1_, the authors suggested that a decreased expression of RAS components could lead to a reduced wall thickness and vascular remodeling which may result in cerebral aneurysms [37].

Preclinical data do not completely support these observations. In an experimental model of aneurysm in rats combining a left internal carotid artery ligation with a left renal artery ligation and a high salt diet (to increase blood pressure), no change in AT_1_ and ACE protein expression could be observed in cerebral aneurysmal walls. The authors therefore excluded any implication of AT_1_ or ACE in cerebral aneurysm formation [38,39], as further supported by the lack of effect of an ARB treatment [38]. If not involved in the development of cerebral aneurysms, RAS has been proposed to play a pivotal role in their rupture. In a mouse model of cerebral aneurysm (obtained by combining induced systemic hypertension with a single injection of elastase into the cerebrospinal fluid at the right basal cistern), Tada et al. prevented the spontaneous aneurysmal rupture with either an ACE inhibitor or an ARB, in a pressure-independent manner [40]. It is noteworthy that, unlike the rat model evoked above, this particular model of intracranial aneurysm showed an increased expression of Ang II and AT_1_ in aneurysmal walls, as revealed by immunohistochemistry [40].

The role of AT_1_ in the pathogenesis of cerebral aneurysm still appears to be controversial. Even if some data suggest a less consistent role of AT_1_ compared to other cerebrovascular diseases, further investigations are required to clarify the role of AT_1_ and its alternative signaling pathways in cerebral aneurysm.

### 2.3. Role of AT_1_ Activation in Traumatic Brain Injury

The Ang II/AT_1_ axis also plays a pivotal role in traumatic brain injury (TBI), another major cause of death and disability. A direct lesion of the brain will result in neuronal damage, increased cell death, dysfunction of the blood-brain barrier and increased inflammatory response. TBI may thus lead to motor and cognitive disorders. In victims of severe TBI, the presence of the D allele of ACE genotype, leading to increased ACE activity, has been correlated with disease mortality [41]. The subsequent excessive AT_1_ activation might contribute to neuronal injury and vulnerability by inducing cerebrovascular remodeling and inflammation [42].

Two genes encode for AT_1_ in rodents, AT_1A_ and AT_1B_, and both share 93–94% amino-acids identity with the unique human AT_1_ protein [43,44]. AT_1A_ has been described as the most abundant AT_1_ isoform expressed in mouse brain [45]. A decrease in susceptibility to traumatic injury was observed in AT_1A_ knockout mice [46].

### 2.4. Role of AT_1_ Activation in Neurodegenerative Diseases

AT_1_ has also been reported as key determinant in many neurodegenerative diseases such as Alzheimer’s and Parkinson’s diseases. However, the involvement of AT_1_ in these pathologies appears to be mostly located at the neuronal level. Recent reviews have been published on this matter [19,20,47] and this will not be further detailed here, as our present review addresses the cerebrovascular renin–angiotensin system.

### 2.5. Limitations

It is noteworthy that most of the studies cited above evaluated AT_1_ protein expression by means of antibodies targeting AT_1_ (Western blot and/or immunohistochemistry). These antibodies are notoriously unspecific [48,49,50] and the results obtained with these antibodies have therefore to be interpreted with caution. Nevertheless, the involvement of AT_1_ in cerebrovascular diseases has been proven by many other means.

The roles of AT_1_ in cerebrovascular diseases are summarized in Figure 1.

## 3. Blocking AT_1_ by ARBs in Cerebrovascular Disease: Success and Limitations

AT_1_ thus appears to be a first-order therapeutic target against cerebrovascular diseases, as has previously been clearly established for cardiovascular diseases. Currently, eight ARBs are clinically available, among which the most studied are losartan, candesartan, telmisartan and valsartan (Figure 2). Often described as surmountable (losartan) or unsurmountable (candesartan, telmisartan, valsartan) antagonists on the basis of their impact on the Ang II concentration response curve [51], ARBs show an inverse agonistic action by stabilizing the inactive state of AT_1_ (see Section 4.2 below). ACE inhibitors and ARBs have been extensively studied to assess their potential preventive and curative effects in cerebrovascular pathologies.

Numerous preclinical studies show the beneficial effects of ARBs at the cerebrovascular level. ARBs regulate cerebral blood flow by blocking AT_1_ in cerebral arterial VSMC and endothelial cells [52]. ARBs improve blood perfusion, reduce apoptosis and decrease the infarct size after an ischemic stroke [53]. A 2-week pretreatment with candesartan before MCAo ischemic strokes in normotensive or hypertensive rats reduces the infarct volume and improves the lower and upper limits of cerebral blood flow autoregulation [54]. This chronic treatment also reduces vascular remodeling [55] and neuronal damage after cerebral ischemia [56,57]. Candesartan has also been reported to reduce neuronal damage in mice after a TBI [58] while valsartan could prevent these lesions [59]. Losartan has shown beneficial effects to prevent aneurysmal subarachnoid hemorrhage in a murine model [40]. Taken together, these preclinical data (which have recently been extensively reviewed [33,60]) provide evidence for the cerebrovascular potential of ARBs.

However, the transposition to clinic does not entirely support these preclinical data. Numerous clinical studies highlight the beneficial effect of long-term RAS blockades on stroke prevention. Early trials were conducted with ACE inhibitors. The stroke rate was reduced by 25% in patients receiving the ACE inhibitor captopril in the CAPPP (CAPtopril Prevention Project) trial [61] and this reduction reached 32% in the HOPE (Heart Outcome Prevention Evaluation) trial with the ACE inhibitor ramipril [62]. Similar drops in stroke recurrence were recorded in patients treated with perindopril, another ACE inhibitor, in the PROGRESS (perindopril PROtection aGainst REcurrent Stroke Study) study [63]. Interestingly, clinical studies using ARBs show identical results, suggesting that the beneficial effect of ACE inhibitors is mainly related to the prevention of AT_1_ stimulation. The LIFE (Losartan Intervention For Endpoint reduction in hypertension) study demonstrated that treatment with losartan prevented the risk of ischemia by 25% as compared to β-blockers that target β-adrenergic receptors [64]. The impact of candesartan was examined in the SCOPE (Study on COgnition and Prognosis in the Elderly) trial in which a risk reduction of 28% for non-fatal stroke was observed [65]. Similar results were found in the MOSES (MOrbidity and mortality after Stroke, Eprosartan compared with the calcium channel blocker nitrendipine for Secondary prevention) study with a 24% reduction in the stroke incidence in people treated with eprosartan [66]. Even if some studies failed to show such a dramatic effect (e.g., TRANSCEND [67] and PROFESS [68] trials), the prevention of strokes afforded by chronic ARBs treatment is well established. However, their efficacy in the acute and post-acute phases of strokes remains a matter of debate. The ACCESS (Acute Candesartan Cilexetil therapy in Stroke Survivors) study, in which candesartan therapy was initiated within 36 h after a stroke, revealed a lower mortality and a lower rate of vascular events after a one-year follow-up [69]. But a more recent trial failed to confirm these results. In the SCAST (Scandinavian Candesartan Acute Stroke Trial) study, candesartan treatment administered within 30 h after a stroke failed to show any beneficial effect when compared to placebo after 6 months [67,70]. In addition, ARBs act neither on cerebral lesions nor on the reduction in infarct volume, while this was observed in preclinical models. Thus, when considering acute stroke treatment, more in-depth studies should be carried out and the search for other therapeutic avenues that target AT_1_ should be considered.

## 4. Alternative Ligands to Target AT_1_ Signaling

AT_1_ has been a matter of intensive research over the last decades and new mechanisms for altering AT_1_ signaling have recently emerged, related to the particular properties of this GPCR (Figure 3).

### 4.1. The Classical View of AT_1_ Signaling

After Ang II binding, AT_1_ adopts an active conformation that stimulates the G_q/11_/phospholipase Cβ cascade [71] to induce VSMC contraction. Indeed, this pathway generates second messengers by phosphatidylinositol-4,5-diphosphate (PIP_2_) hydrolysis. The increase in inositol trisphosphate (IP_3_) will result in myosin light chain kinase (MLCK) activation through Ca^2+^/calmodulin complexes. The increase in intracellular Ca^2+^ has also been reported to activate the JAK2 kinase which phosphorylates the ARHGEF1 Rho guanine nucleotide exchange factor [72]. ARHGEF1 then activates the RhoA/Rho kinase–mediated phosphorylation of MYPT1 (myosin phosphatase target subunit 1) and inhibition of the myosin light chain phosphatase (MLCP) activity [72].

Recent studies have also reported that AT_1_ coupling to G_i/o_, G_12_ and G_13_ [5,6,73,74,75,76]. Coupling of AT_1_ to G_i/o_ leads to an inhibition of adenylyl-cyclase and a decrease in cyclic adenosine monophosphate in kidneys, adrenal glomerula zone or liver [6]. AT_1_ activation, via its coupling to G_12/13_, may also trigger the RhoA/Rho kinase pathway evoked above and thus participate to vascular contraction, remodeling and inflammation induced by Ang II [77]. AT_1_ has also been reported to phosphorylate and activate the cytosolic phospholipase PLA2 [78], resulting in the release of arachidonic acid and its derivatives which are involved in vascular tone and in p22phox-based NADH/NADPH oxidase activation in VSMC [79]. Intracellular protein kinases, such as protein kinase C (PKC), mitogen-activated protein kinases (MAPK) and protein kinase B (AKT), may also stimulate NADPH oxidase and lead to the generation of reactive oxygen species (ROS) induced by Ang II [5,80].

After Ang II stimulation, AT_1_ also undergoes phosphorylation on its C-terminal serine and threonine residues by GPCR kinases (GRKs). This phosphorylation process is a common feature of GPCR and increases β-arrestin1 and β-arrestin2 affinities for the receptor [81,82]. The initial role of β-arrestins was described to arrest the response by interrupting the heterotrimeric G protein coupling and to induce receptor internalization through clathrin-coated pits. But a new role of β-arrestins has now emerged for GPCR [83,84] including AT_1_: once stably recruited by AT_1_, β-arrestins act as scaffolding proteins that are suggested to directly promote AT_1_ alternative signaling cascades with potential protective effects.

### 4.2. Targeting Different Active Conformations of AT_1_ with Biased Agonists

Many ligands of GPCR can selectively initiate one or more distinct signaling pathways when coupled to their receptor. This is the biased agonism theory, also known as functional selectivity [85], that was first described more than four decades ago but not explored until recently [86]. These biased agonists will preferentially stabilize one of the many conformations that GPCR can adopt, thus allowing only specific interactions between the receptor and specific downstream intracellular proteins. In addition to the receptor conformation, cellular background and simultaneous expression of other receptors may lead to phosphorylation changes at the C-terminus of activated receptors. This can induce the binding of different conformations of β-arrestins, different internalization routes and altered signaling. It is referred to as the barcode theory [87,88].

Because the biased signaling exploration held great promise for the development of more effective therapies, the concept became rapidly fashionable and over-simplified, stating that for each GPCR, in addition to the classical active state that couples to both G protein and beta-arrestin, one could find biased ligands triggering exclusive signaling either towards heterotrimeric G coupling or towards β-arrestin. This view of exclusive G proteins *versus* β-arrestins biased signaling has now been debated in a series of valuable articles [89,90,91,92] but still holds for AT_1_.

In the case of AT_1_, several biased agonists have been developed and pharmacologically characterized that allow specific activation of the G protein-dependent pathway without activating β-arrestins (TRV055 and TRV056, Figure 3) and vice versa (TRV023 and TRV027, Figure 3) [93]. Biased ligands that differentially activate AT_1_ coupling among the various G protein subtypes have also been described [75]. Combining very precise molecular and pharmacological characterizations of these biased AT_1_ agonists (allowed by the development of new tools, such as various biosensors based on bioluminescence resonance energy transfer BRET [75]) together with structural information [94,95,96] and molecular dynamic simulations of different categories of biased ligands/AT_1_ pairs [71,97,98], AT_1_ is now demonstrated to adopt at least three conformational states: the inactive one, the canonical active conformation that can couple to both G_q_ and β-arrestin pathways, but also an alternative active conformation coupling almost exclusively to β-arrestins. The following paragraph describes at the molecular level the results arising from these recent and highly relevant studies.

The very first X-ray crystallography study of AT_1_ consisted of the receptor co-crystallized with an ARB [99]. The resulting inactive form of the receptor showed an H-bond between N295 of the transmembrane domain TM7 (N295^7.46^) and N111 of the TM3 (N111^3.35^). This interaction ensures an inward positioning of the TM6, together with a slight outward movement of TM7, precluding any contact with intracellular effectors (Figure 4, in salmon). This conformation is stabilized by biphenyl-tetrazole ARBs, as they tightly interact with several residues, including N295^7.46^. They afford a stabilization of the N111^3.35^–N295^7.46^ H-bond and of the network of surrounding residues in the inactive conformation [51,99]. This stabilization of AT_1_ in an inactive state results in the pharmacologically observed robust inverse agonism.

In contrast, a more recent X-ray crystallography study [98] showed that in the presence of Ang II or of the biased agonist TRV 023 (Figure 4, in green and cyan, respectively), interaction with agonists induces an outward shift of TM6, which is a hallmark of GPCR activation [96]. In AT_1_, this shift in TM6 is due to a conformational change of N295^7.46^. Other events depend on the nature of the agonist, whether biased or not and have been well described by Suomivuori et al. and Wingler et al. in two recent and complementary articles published in 2020 [96,98]. The authors could decipher the following mechanisms by molecular dynamics and X-ray crystallography, both techniques providing converging results which are given below.

The natural agonist, Ang II, inserts itself in the ligand pocket of the receptor with its N-terminus oriented toward the extracellular side. The C-terminus with its hydrophobic phenylalanine will therefore position itself deeply into the core of the receptor, near the hydrophobic L112 of TM3 (L112^3.36^) and pushing it towards TM2. Consequently, the neighboring N111^3.35^ points outward from the receptor (Figure 5a in green). As a result, instead of pointing toward TM3, Y292^7.43^ points toward TM2, exchanging its position with F77^2.53^. This movement of Y292^7.43^ induces a clockwise twist of TM7, such that the N46^1.50^ below establishes an H-bond with N295^7.46^ instead of C296^7.47^ [98]. It also causes an outward shift of TM7 in the intracellular part of the receptor permitting Y302^7.53^ to point up (Figure 5b in green). Such a global phenomenon, described by Suomivuori et al. as a “long-range allosteric network”, drives the information from the ligand-binding pocket to the intracellular side of AT_1_, resulting in the “canonical active conformation”. The same pattern occurs with a greater extent when the ligand is a G_q_ biased agonist, allowing sufficient space for the α5 helix of G_q_ to accommodate with the receptor.

Alternatively, most arrestin-biased peptide agonists do not possess a phenylalanine on their C-terminus (Figure 6a) [94,100]. The consecutive lack of hydrophobic interaction with L112^3.36^ prevents the complete outward shift of the intracellular side of TM7, leading to the “alternative active conformation” (Figure 6b in cyan). It includes a downward rotamer of Y302^7.53^ which then points to the intracellular side. This has a repelling effect on R126^3.50^, thus also pointing down (Figure 5b, in cyan). R126^3.50^ belongs to the conserved DRY motif of GPCR that is mandatory for GPCR coupling to heterotrimeric G proteins. The models simulated by Suomivuori et al. have established that this downward position of R126^3.50^ induces a steric clash with the α5 helix of G_q_, preventing its coupling to AT_1_ [98]. The same models have shown that both AT_1_ active conformations (canonical and alternative) can accommodate with the β-arrestin finger loop [98].

A very seductive hypothesis has been proposed to explain why AT_1_ can adopt such an alternative active conformation that binds almost exclusively to β-arrestin and not G proteins [96]. Wingler et al. have suggested that a biased signaling switch may predispose AT_1_ to adopt an exclusive β-arrestin conformation. Interestingly, Ang II binding to AT_1_ not only induces changes at the bottom of the ligand-binding pocket but also in residues constituting a key polar network in the transmembrane region of the receptor. In most class A GPCR, this polar network forms a sodium-binding site stabilizing the inactive conformation. This Na^+^/GPCR interaction is due to the bond established between Na^+^ and a highly conserved serine in TM7. However, one of the specificities of AT_1_ among GPCR family, is that this serine residue is replaced by the asparagine N295^7.46^ (involved in the initial outward shift of TM6 evoked above), thus alleviating Na^+^ binding and favoring the β-arrestin alternative active conformation. Similar residue substitutions in the conserved Na^+^-binding motif have been reported in several atypical chemokine receptors, such as the decoy receptor CXCR7 [101,102], that do not signal through G protein coupling, strengthening the hypothesis that the absence of the Na^+^ binding site could favor an exclusive β-arrestin active conformation.

### 4.3. Targeting AT_1_ by Covalent S-Nitrosation

*S*-nitrosation is defined as the covalent bond between nitric oxide (NO, a well-known gaseous transmitter playing an essential role in cardiovascular homeostasis) and the thiol moiety of cysteine residues of target proteins [103]. The so formed *S*-nitrosothiols are able to induce trans-nitrosation (the transfer of NO to another target protein) and/or to release NO, thus conferring on them potent vasodilator properties [104]. The *S*-nitrosation of proteins can induce dramatic changes in their functions [105] and recent papers suggest that AT_1_ functions could be regulated by *S*-nitrosation.

Studies carried out by Leclerc et al. in 2006 showed a slight decrease in the affinity of Ang II for the AT_1_ following exposure to sodium nitroprusside (SNP), a NO donor, in a cellular model of HEK293 cells overexpressing AT_1_ [106]. Interestingly, it was proposed that AT_1_ is the direct target of NO: using several AT_1_ cysteine mutants, the authors showed that mutation of the cysteine 289 (C289^7.40^) for a serine in human AT_1_ was no longer sensitive to NO, suggesting that *S*-nitrosation of the cysteine residue at position 289 in TM7 of AT_1_ was responsible for the slight loss of affinity for Ang II [106].

Based on these results, our group proposed that NO donors could directly reduce arterial contraction induced by Ang II activation of AT_1_. We investigated the physiological consequences of this phenomenon by studying the effect of NO in an ex-vivo model: isolated and perfused middle cerebral arteries of normotensive rats [107]. *S*-nitrosation was induced by pretreatment with *S*-nitrosoglutathione (GSNO), an endogenous *S*-nitrosothiol previously shown as protective in different models of cerebral ischemia (MCAo) [108,109] and autologous blood clots injected through the internal carotid artery [110]). GSNO pretreatment of rat middle cerebral arteries specifically abolished the vasoconstriction induced by Ang II. Other vasoconstrictors sharing the same G_q_ signaling pathway, such as phenylephrine, or targeting Gi/o coupled receptors, such as serotonin, still produced vasoconstriction after GSNO pre-treatment. GSNO therefore specifically inhibited the vasoconstriction induced by Ang II by acting upstream of the G_q_ proteins, i.e. probably on the receptor itself. In preliminary experiments, we could detect *S*-nitrosated AT_1_ after exposure of HEK293 cells overexpressing AT_1_ to GSNO, although we did not investigate which cysteine was targeted [107]. Similar results have just been published [111]. Using an ex-vivo model of rat aortic rings, Pinheiro et al. have observed a decrease in vascular contraction induced by Ang II when arteries were pretreated with nitrite or GSNO [111]. In vivo, a decrease in the blood pressure response to Ang II was also detected after oral administration of nitrite or GSNO [111]. Both treatments increased plasma concentrations of *S*-nitrosated species and enhanced total protein *S*-nitrosation in the vessels. The authors identified PKC as the target of *S*-nitrosation and as the underlying mechanisms of the decreased responses to Ang II. In Leclerc’s study, a decrease in the EC50 of Ang II-induced inositol phosphate production upon SNP treatment was reported, while Pinheiro et al. found, in HEK293T cells expressing AT_1_, a Bmax decrease in the Ang II-induced intracellular calcium mobilization after GSNO pre-treatment. Taken together, these studies suggest that *S*-nitrosation promoted by GSNO reduces the physiological responses induced by Ang II by acting directly on AT_1_ and on other proteins such as PKC.

Interestingly, the slight decrease in Ang II affinity described by Leclerc et al. after AT_1_ *S*-nitrosation might not be the only effect of NO on the receptor. Indeed, the AT_1_ dependent myogenic tone (the arterial contractile response consecutive to an increase in arterial transmural pressure and mechanical stretch [112]), which is independent of the binding of Ang II to AT_1_, decreases in isolated middle cerebral arteries pretreated with GSNO (Figure 3) [107]. In addition, in preliminary experiments, GSNO pre-treatment of HEK293 cells did not impair the kinetics and amplitude of activated AT_1_ internalization [107]. This suggests that the AT_1_/β-arrestin pathway could remain unaffected by the *S*-nitrosation of the receptor.

When looking at AT_1_ structure, it is noteworthy that C289^7.40^ is one of the residues involved in the “long-range allosteric network” (see Section 4.2). As shown in Figure 7, the thiol function of C289^7.40^ is oriented inside the receptor in the alternative active conformation (in cyan), whereas it points outward from the receptor in the inactive (in salmon) and canonical active (in green) states. This suggests that any modification on this particular residue could alter the process involved in conformational changes and may preclude or stabilize one or more conformations of AT_1_. This hypothesis is strengthened by several reports focusing on single nucleotide polymorphism of AT_1_ in humans. Among the few naturally occurring variants of AT_1_, one is related to C289^7.40^ where the cysteine is replaced by a tryptophan residue (C289W variant). This variant has not been associated with a pathology, but seems to be less expressed at the plasma membrane and decreases the potency of Ang II on phosphatidylinositol hydrolysis consecutive to G_q_ stimulation [113]. At least two other recent studies confirmed this reduced affinity of the C289W variant for Ang II but also showed that C289W is able to interact with β-arrestin2 at the plasma membrane upon Ang II stimulation [75,114]. However, C289W alters the β-arrestin2 conformation and reduces the β-arrestin-dependent activation of ERK_1/2_ [114]. Taken together, these data support the idea that a modification at C289^7.40^, such as *S*-nitrosation, may bias AT_1_ signaling, but further investigations are needed.

## 5. Promoting the β-Arrestin Pathway of AT_1_ to Treat Cerebrovascular Disease

Having established that AT_1_ is prone to biased agonism towards the β-arrestin pathway, we will now review the potential interests of promoting this AT_1_/β-arrestin signaling. AT_1_ recruits both β-arrestins equally and internalizes as a stable complex, inducing a slow recycling to the plasma membrane [115,116].

### 5.1. Biased Agonists towards AT_1_/β-Arrestins in Cardiovascular Diseases

In addition to the desensitization and internalization of AT_1_ which reduce G_q_ signaling, the activation of the β-arrestin pathway induces several signaling cascades involved in cellular processes, such as cell proliferation and growth, nutrient uptake and migration (Figure 3). Following stimulation either by Ang II or by the Ang II peptide analog SII, the first β-arrestin biased AT_1_ agonist identified (Figure 6a), the consecutive phosphoproteomes do not overlap, suggesting a G_q_-independent phosphorylation process induced by β-arrestin biased agonists [117,118]. Moreover, in a very recent proteomic study, Pfeiffer et al. showed that the dynamic recruitment of proteins in close proximity to AT_1_ displays a completely different pattern as a function of time, depending on whether AT_1_ was stimulated by full agonists, G_q_ biased agonists, or β-arrestin biased agonists [119].

As previously mentioned, both β-arrestin1 and β-arrestin2 can be recruited after the phosphorylation of AT_1_ by GRKs. The use of β-arrestin1 and β-arrestin2 biosensors has shown that a given ligand/AT_1_ pair recruits both β-arrestins to the same extent but that they stabilize the receptor in distinct conformations [120]. Indeed, although the primary sequence and 3D structure of the two β-arrestin isoforms are very similar, conformational differences have been reported, suggesting functional differences [121]. This is illustrated at the cardiovascular level: on the one hand, β-arrestin1 exhibits harmful effects at the cardiac level, in particular by reducing cardiac function, promoting apoptosis and inflammation while, on the other hand, β-arrestin2 has been shown to have protective cardiac effects [122]. β-arrestin2 mediates signaling through Src, ERK_1/2_, AKT and PI3K activations and has been reported to be cytoprotective, anti-apoptotic and pro-survival [93,123,124,125,126]. β-arrestin2 has also been reported to inhibit the inflammation process after myocardial infarction [127]. However, one study has shown conflicting results, showing that β-arrestin2 is involved in cell death signaling after ischemic cardiac injury [128].

The importance of the exclusive AT_1_/β-arrestin pathway at the cardiovascular level has been studied through the use of various ligands biased towards β-arrestins, such as TRV027 and TRV023 (Figure 3 and Figure 6a) [129,130] and has been recently reviewed by Turu et al. [92]. Only TRV027 has been studied in clinical trials for heart failure (NCT01187836), kidney disease (NCT01444872) and currently for COVID-19/SARS-CoV-2 infection (NCT04419610; NCT04924660).

TRV027 increases cardiac contractility in vitro and reduces blood pressure in rats [131]. In addition, in a canine model of cardiac insufficiency, TRV027 reduces systemic and renovascular resistance and promotes cardiac unloading [132,133]. Based on these promising results, TRV027 has also been tested in humans, both in healthy subjects [134] and in patients with acute heart failure in the BLAST-AHF (Biased Ligand of the Angiotensin receptor STudy in Acute Heart Failure) study [135]. The beneficial effects of TRV027 were not confirmed by this first clinical study [136]. However, the interest of such β-arrestin-biased agonists of AT_1_ remains, since many other ligands with similar bias and convincing preclinical data are still to be tested.

Hence, TRV023 is another AT_1_ agonist that blocks G protein signaling and promotes the recruitment of β-arrestins (Figure 6a). In mice, TRV023 increases cardiac contractility in a β-arrestin2-dependent manner while it decreases blood pressure independently of β-arrestin2 (most likely through inhibition of the G_q_ pathway) [130]. TRV023 reduces cell death both ex vivo (intraventricular balloon-induced membrane stretch injury) and in vivo (left ventricular ischemia-reperfusion injury) by activating the MAPK and AKT pathways. These cytoprotective effects were not observed with losartan or in β-arrestin2 KO mice [130]. These results are strengthened by previous works showing that mechanical stretching in cardiac myocytes facilitates β-arrestin2 biased pro-survival signaling via EGFR transactivation mediated by AT_1_ independently of G protein or Ang II [137]. Moreover, TRV023 prevents Ang II-induced cardiac hypertrophy while preserving contractility more efficiently than losartan by improving the calcium response of myofilaments [138]. Finally, TRV023 also induces an increase in myocardiac contractility and in ventricular myosin light chain-2 phosphorylation in a transgenic mouse model of familial dilated cardiomyopathy [139].

Thus, the beneficial actions of these β-arrestin biased AT_1_ ligands appear promising, although not yet fully understood. These biased agonists also seem to bind to AT_2_, which could contribute to counteracting the cardiovascular harmful effects of AT_1_. Although AT_2_ does not couple to heterotrimeric G proteins and does not recruit β-arrestins [76,140], this receptor may play a role in the in vivo effects of these biased agonists, e.g., by modulating AT_1_/AT_2_ heterodimer signaling. However, to date no comprehensive study has been conducted to support this hypothesis.

### 5.2. Biased Agonists towards AT_1_/β-Arrestins in Cerebrovascular Diseases

To the best of our knowledge, no study has focused on the impact of specific β-arrestin biased agonists of AT_1_ on cerebrovascular diseases. The literature, however, supports the idea that these drugs might represent new hope for these clinical situations in which alternative treatments are lacking.

Interestingly, Ang-(1–7), a major component of the Ang-(1–7)/MasR/ACE2 protective axis of the renin–angiotensin system, has also been described as a β-arrestin biased ligand of AT_1_ [74,141]. Numerous studies report cardioprotective effects of Ang-(1–7) [141,142,143,144] as well as cerebroprotective effects in ischemic and hemorrhagic strokes [60,145,146]. Ang-(1–7) protects against central nervous system damage and neurological deficits induced by experimental ischemic strokes [145]. It reduces the infarct volume, oxidative stress, as well as the release of pro-inflammatory cytokines through Mas receptor stimulation [147]. However, even if MasR antagonists were used to prevent Ang-(1–7) effects, this does not completely rule out that its weak, exclusive β-arrestin biased agonist properties on AT_1_, together with its AT_2_ agonistic properties might contribute to the cerebrovascular beneficial effects of Ang-(1–7) [76,140,148,149]. The heterodimeric complexes between AT_1_/AT_2_ or AT_1_/MasR seem to inhibit AT_1_ signaling by Ang II, β-arrestin recruitment and receptor internalization [150,151,152,153]. It would be interesting to investigate how Ang-(1–7) and other β-arrestin biased ligands affect the signaling of these heterodimers.

Unlike reports at the cardiac level, both β-arrestin1 and β-arrestin2 seem to have neuroprotective effects on cerebrovascular diseases. The MCAo ischemic stroke model in mouse revealed an increase in protein expression of the two β-arrestins, suggesting their involvement in the pathophysiological process [154]. This study also reported that β-arrestin1 deletion worsens ischemic brain damage assessed by neuronal deficits, infarct size and mortality. These effects are correlated with the suppression of autophagy and induced neuronal apoptosis and necrosis in vitro and in vivo [154]. These results suggest that β-arrestin1 has neuroprotective effects related to autophagy. However, we should remain cautious, since β-arrestin1 deletion may affects a very wide variety of cellular functions, whether beneficial or deleterious.

Studies exploring the alterations of β-arrestin2 expression in strokes and neurovascular disease are scarce and the role of β-arrestin2 in neurological disorders is not yet clear. One study showed that β-arrestin2 plays a pivotal role in the immunodeficiency syndrome induced by strokes [155]. In particular, β-arrestin2 is believed to be a key intracellular mediator of adrenergic activity in monocytes after a stroke and may therefore be implicated in the stroke-induced adrenergic-mediated monocyte dysfunction [155]. However, these in vitro results concluding that there is a harmful role of β-arrestin2 after a stroke are not supported by in vivo experiments. Indeed, a more recent study suggests that β-arrestin2 signaling improves endothelial function and blood–brain barrier integrity after a stroke [156]. Using APC (activated protein C), a β-arrestin2 biased agonist of the PAR-1 receptor (protease-activated receptor 1), the authors showed that β-arrestin2 signaling has protective effects in mice under ischemic conditions (MCAo), notably as it increases angiogenesis. A high fat diet reduced β-arrestin2 expression in mice and the β-arrestin2 levels in the penumbra were inversely correlated with the severity of ischemic lesions after MCAo [156]. The authors identified PDFG-β as one of the key target molecules activated by β-arrestin2 to promote angiogenesis [156].

## 6. Conclusions

The pathophysiological role of AT_1_ in cerebrovascular diseases has been increasingly highlighted in recent years; however, therapeutic advances in this area have not followed. The development of new therapeutic approaches requires a better understanding of AT_1_ signaling pathways and its possible interactions with other components. Since β-arrestin-biased AT_1_ agonists have showed promising results in the cardiovascular disease context, it would be interesting to study them in models of cerebrovascular diseases such as strokes.

There is still much to learn about the regulation of AT_1_ signaling, whether through β-arrestin signaling or characterization of the potential beneficial effects of β-arrestin biased agonists.

## Figures and Tables

**Figure 1 ijms-22-06738-f001:**
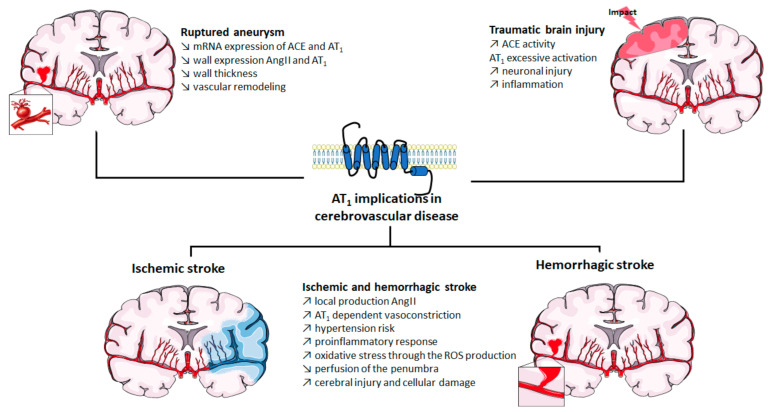
Implication of AT_1_ in cerebrovascular diseases. ACE: angiotensin-converting enzyme; Ang II: angiotensin II; AT_1_: angiotensin II type 1 receptor; mRNA: messenger ribonucleic acid; ROS: reactive oxygen species. The figure was prepared using the Servier Medical Art website (Creative Commons Attribution 3.0 unported License).

**Figure 2 ijms-22-06738-f002:**
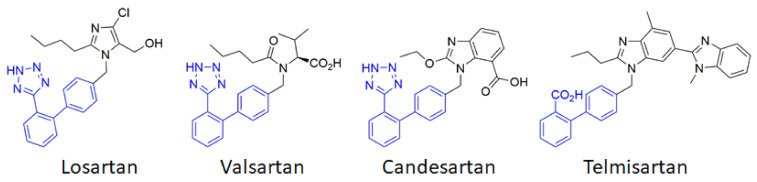
Chemical structures of the main angiotensin receptor blockers (ARB) used in therapeutics. The biphenyltetrazole (or the bioisosteric biphenyl carboxylic acid for telmisartan) is highlighted in blue.

**Figure 3 ijms-22-06738-f003:**
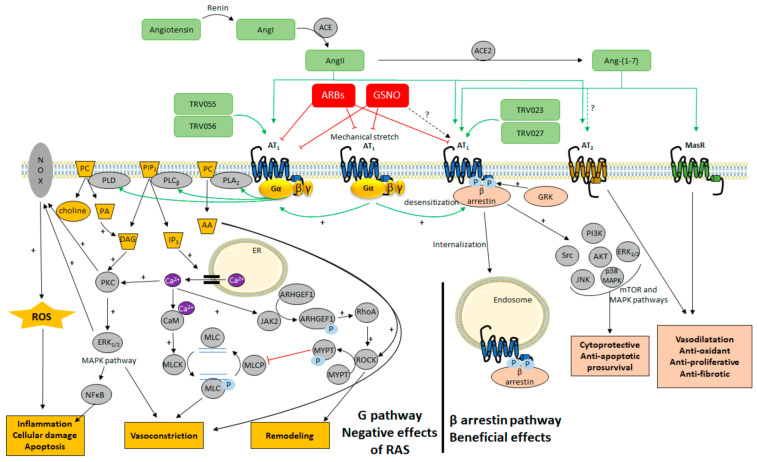
Overview of the cellular signaling pathways activated by AT_1_ after stimulation of the renin–angiotensin system (RAS). AA: arachidonic acid; ACE: angiotensin-converting enzyme; ACE2: angiotensin-converting enzyme 2; AKT: protein kinase B; Ang-(1–7): angiotensin (1–7); Ang I: angiotensin I; Ang II: angiotensin II; ARHGEF1: Rho guanine nucleotide exchange factor 1. AT_1_: angiotensin II type 1 receptor; AT_2_: angiotensin II type 2 receptor; ARBs: angiotensin receptor blockers; CaM: calmodulin; DAG: diacylglycerol; ER: endoplasmic reticulum; ERK_1/2_: extracellular signal regulated kinase ½; GRK: G protein-coupled receptor kinase; GSNO: S-nitrosoglutathione; IP_3_: inositol triphosphate; JAK2: Janus kinase 2; JNK: c-Jun N-terminal Kinase; MAPK: mitogen-activated protein kinases; MasR: Mas receptor; MLC: myosin light chain; MLCK: myosin light chain kinase; MLCP: myosin light chain phosphatase; mTOR: mammalian target of rapamycin; MYPT: myosin phosphatase target subunit; NFκB: nuclear factor kappa B; NOX: NADPH oxidase; PA: phosphatidic acid; PC: phosphatidylcholine; PI3K: phosphoinositide 3 kinase; PIP_2_: phosphatidylinositol-4, 5-diphosphate; PLA_2_: phospholipase A2; PLC_β_: phospholipase Cβ; PLD: phospholipase D; PKC: protein kinase C; RhoA: RAS homolog family member A; ROCK: Rho associated protein kinase; ROS: reactive oxygen species; TRV023 and TRV027: β-arrestin biased AT_1_ agonists; TRV055 and TRV056: G_q_ biased AT_1_ agonists. The figure was prepared in part with the Servier Medical Art website (Creative Commons Attribution 3.0 unported License).

**Figure 4 ijms-22-06738-f004:**
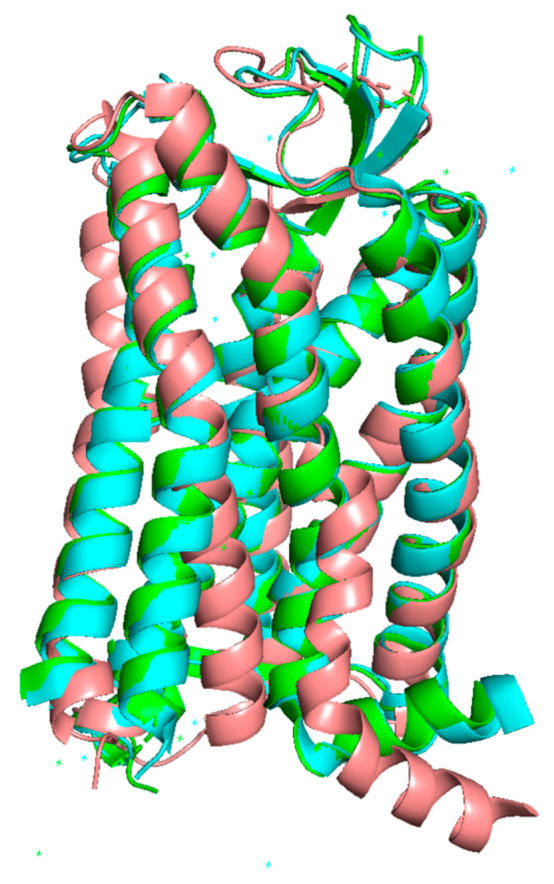
Superimposition of the three known structures of AT_1_. Inactive (salmon), canonical active (green) and alternative active (cyan). TM6 and TM7 are located in the front. In the inactive structure, TM6 is remarkably bent inside the receptor, while TM7 is slightly twisted outside. The reference inactive structure of AT_1_ has been chosen as PDB code 4YAY (from [99]). Canonical active and alternative active structures have been chosen as PDB code 6OS0 and 6OS1 respectively (from [96]). The figure was prepared with the PyMOL software (The PyMOL Molecular Graphics System, Version 2.5.0. Schrödinger, LLC).

**Figure 5 ijms-22-06738-f005:**
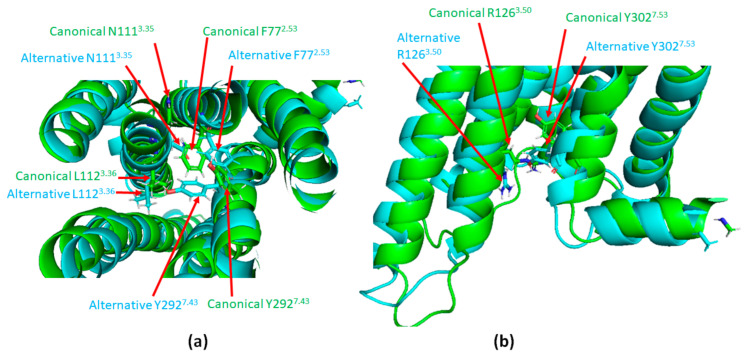
Illustration of the long range allosteric network of AT_1_ inducing its switch to canonical active conformation. (**a**): Top view of the canonical (green) and alternative (cyan) active conformations of AT_1_. (**b**): Superimposition of the intracellular part of AT_1_ in its canonical (green) and alternative (cyan) active conformations. Canonical active and alternative active structures are simulation data available in supplementary material from [98]. The figure was prepared with the PyMOL software (The PyMOL Molecular Graphics System, Version 2.5.0. Schrödinger, LLC).

**Figure 6 ijms-22-06738-f006:**
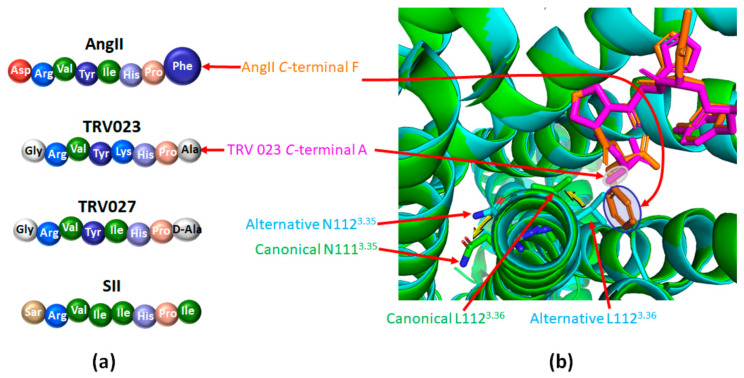
β-arrestin biased agonists of AT_1_ favor its alternative active conformation. (**a**): Comparison of the amino acid sequences of angiotensin II (Ang II) and of the main β-arrestin biased agonists of AT_1_; Ang II: angiotensin II; SII: [Sar^1^,Ile^4^,Ile^8^] angiotensin II peptide analog; TRV023 and TRV027: β-arrestin biased AT_1_ agonists. (**b**): Superimposition of the structure of AT_1_ bound to Ang II (green, PDB code 6OS0; Ang II appears in orange) or to the β-arrestin biased agonist TRV023 (cyan, PDB code 6OS1; TRV023 appears in purple). The phenyl group of the C-terminal part of Ang II (highlighted in blue) goes deep into the ligand pocket, inducing a rotation of L112^3.36^ and of N111^3.35^ (yellow arrows). It yields the canonical structure of AT_1_ (appears in green). The C-terminal part of TRV023 comprises a less bulky alanine residue (highlighted in grey) which does not induce such motion, yielding the alternative structure (in cyan). This panel was prepared with the PyMOL software (The PyMOL Molecular Graphics System, Version 2.5.0. Schrödinger, LLC).

**Figure 7 ijms-22-06738-f007:**
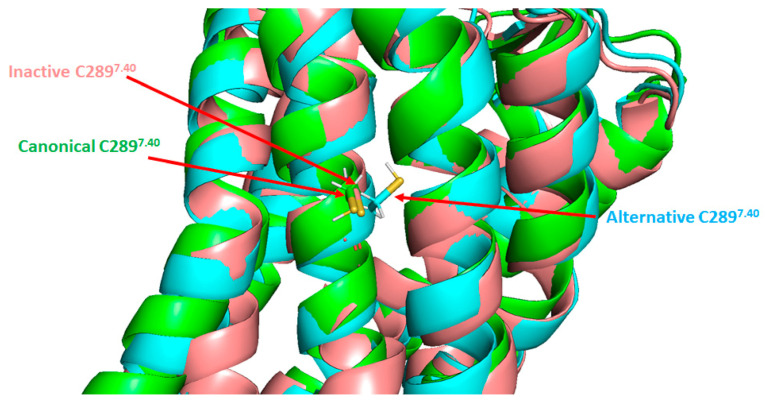
Positions of the cysteine 289 residue in the three known AT_1_ structures. Superimposition of the three known AT_1_ structures: inactive (salmon), canonical active (green) and alternative active (cyan). TM6 and TM7 are located in the front. C289 residue is indicated by arrows. The reference inactive structure of AT_1_ has been chosen as PDB code 4YAY (from Zhang et al. [99]). Canonical active and alternative active structures are simulation data available in supplementary material from Suomivuori et al. [98]. The figure was prepared with the PyMOL software (The PyMOL Molecular Graphics System, Version 2.5.0. Schrödinger, LLC).

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
