# Peer review of "Targeting the Angiotensin II Type 1 Receptor in Cerebrovascular Diseases: Biased Signaling Raises New Hopes"

_ijms, 2021, doi:10.3390/ijms22136738_

Round 1
Reviewer 1 Report
Authors have elaboratively explained promoting beta-arrestin for cerebrovascular diseases. It is well written.
Minor concern: Authors may consider providing a separate table that has information on modulators of beta-arrestin/if any molecules that underwent clinical trials.
Author Response
We thank the reviewer for his comment on the manuscript.
We did not add a table as requested by the reviewer because only four clinical trials have been (or are being) conducted with β-arrestin-biased agonists of AT1 and only one of them is directly related to cardiovascular pathologies. Instead, we added a sentence citing these 4 clinical trials. We hope this option will satisfy the reviewer.
Reviewer 2 Report
The Review Article:” Interest of promoting the beta-arrestin pathway of angiotensin 2 II type I receptor for cerebrovascular disease” by Céline Delaitre et al. discuss the pathophysiological role of AT1 receptors activity in the Renin-Angiotensin System. The Review describes the problem of biased agonism of ligands activating AT1 has been a fundamental issue in dealing with side effects and finding more effective therapeutic approaches for cerebrovascular diseases. The Article message, however, should be provided clearer. The Review Article is lengthy and contains information not related directly to its aim.
Specific comments:
The Review Article title: “Interest of promoting the beta-arrestin pathway of angiotensin 2 II type I receptor for cerebrovascular disease” does not correspond to the knowledge presented in the manuscript. The Article discusses the engagement of AT1 in cerebrovascular diseases instead of the role of beta-arrestin signaling in pathomechanism/treatment of cerebrovascular diseases. Please, adjust the title of the Review Article to its content.
In the current shape, the introduction section does not prepare the reader to understand the rest of the Article. The body text is lengthy, chaotic, sometimes not related directly to the content of the Article Review. See, e.g., detailed specification of methods for the assessment of amino acids and gene homology). In turn, basic information about RAAS is laconic. Please improve the layout of the content in the introduction section from general to more detailed and prepare the reader for specialist knowledge reviewed later on. Remove information unrelated to the rest of the Article and shorten the descriptions about places where the reader can find the precise expertise in specific problems. Instead, describe in a general but understandable way necessary information about the structure and functioning of RAS.
One of the stated aims: “We will present emerging concept to target AT1” is not precise, and it is hard to understand which part of the Article present this concept. Please rephrase this aim to improve understanding.
Remove redundancy from the body text throughout the manuscript: E.g., #1- sentence in the Verses 70-75: “AT1 is PRIMARILY coupled to the heterotrimeric Gq/11 protein THAT REPRESENTS THE MAIN AT1 signal transduction process…”; e.g., #2 – information in lines 291-306 is not related directly to AT1.
Chapter 5.1: The title of the chapter: “AT1, a GPCR prone to exclusive β-arrestin biased signaling,” does not reflect the content of this part. The reader can find here hypotheses of conformational states of AT1 without clear evidence that the β-arrestin pathway is the main pathway for AT1. Moreover, this chapter has no clarity why authors link information about specific biased ligands of AT1 with the explanation of conformational states of AT1. My recommendation is to rewrite this chapter to convince the reader of your point of view that a specific alternative active conformational state of AT1 activates β-arrestin signaling instead of G-protein signaling.
There is no consistency in the manuscript with the usage of abbreviations. For angiotensin II receptor type I, the authors use “AT1 receptor” but sometimes just “AT1”. The same is in the case “7TM”. It is not clear from the text why authors introduce two kinds of abbreviation for G protein-coupled receptors: 7TM and GPCR. Please improve the usage of acronyms and their consistency throughout the manuscript.
A list of abbreviations could help the understanding of the text.
Verse 47: the statement “(identified as the receptor for SARS-cov2 virus)” is not related to the rest of the manuscript and should be removed.
Please describe the mechanism of “protective arm” more profound via AT2 and MasR, which you mentioned in the introduction.
Verses 58-62: The sentence: “Angiotensin……diseases”. To improve the understanding of this sentence, I highly recommend using more than one sentence here. Moreover, I recommend avoiding the additional information in brackets and using them only if necessary.
Verses 175-175: The purpose of the statement about the lack of specificity of AT1 antibodies given in chapter 3.2. has not been precisely described in the context provided in this chapter the role of AT1 in cerebral aneurysm disease but not in other conditions. If this information is necessary to the Article, it should be moved to a separate section where the limitation of our knowledge is discussed.
English grammar needs to be corrected by a proofreader specializing in scientific English before resubmitting the manuscript.
Round 2
Reviewer 2 Report
The authors addressed all my critical remarks and applied necessary corrections to the current version of the manuscript, which significantly improved its scientific quality. The revised version of the Review Article: “Targeting the angiotensin II type I receptor in cerebrovascular diseases: biased signaling raises new hopes” by Céline Delaitre et al., in my opinion, is worth publication in IJMS. I believe it will be interesting for readers looking for the mechanism of cerebrovascular diseases or interested in general in the role of AT1 signaling in health and disease.
I found only one typo which should be corrected: in verse 41, information in brackets should be precisely completed. Did authors mean >stimulation, etc. < by the phrase “stimulation…)”?